# Field Evaluation of High Modulus Asphalt Concrete Resistance to Low-Temperature Cracking

**DOI:** 10.3390/ma15010369

**Published:** 2022-01-04

**Authors:** Marek Pszczola, Dawid Rys, Mariusz Jaczewski

**Affiliations:** Faculty of Civil and Environmental Engineering, Gdansk University of Technology, 80-233 Gdansk, Poland; mpszczol@pg.edu.pl (M.P.); marjacze@pg.edu.pl (M.J.)

**Keywords:** asphalt mixture, low-temperature cracking, field sections, climatic conditions, Thermal Stress Restrained Specimen Test (TSRST)

## Abstract

High-modulus asphalt concrete has numerous advantages in comparison to conventional asphalt concrete, including increased resistance to permanent deformations and increased pavement fatigue life. However, previous studies have shown that the construction of road pavements with High Modulus Asphalt Concrete (HMAC) may significantly increase the risk of low-temperature cracking. Those observations were the motivation for the research presented in this paper. Four test sections with HMAC used in base and binder courses were evaluated in the study. Field investigations of the number of low-temperature cracks were performed over several years. It was established that the number of new low-temperature cracks is susceptible to many random factors, and the statistical term “reversion to the mean” should be considered. A new factor named Increase in Cracking Index was developed to analyze the resistance of pavement to low-temperature cracking. For all the considered field sections, samples were cut from each asphalt layer, and Thermal Stress Restrained Specimen Tests were performed in the laboratory. Correlations of temperature at failure and cryogenic stresses with the cracking intensity observed in the field were analyzed. The paper provides practical suggestions for pavement designers. When the use of high modulus asphalt concrete is planned for binder course and asphalt base, which may result in lower resistance to low-temperature cracking of pavement than in the case of conventional asphalt concrete, it is advisable to apply a wearing course with improved resistance to low-temperature cracking. Such an approach may compensate for the adverse effects of usage of high modulus asphalt concrete.

## 1. Introduction

### 1.1. Background

High-modulus asphalt concrete (HMAC) has numerous advantages in comparison to conventional asphalt concrete, such as increased resistance to permanent deformations and increased pavement fatigue life [1,2]. However, previous studies [3,4] have shown that the construction of road pavements with HMAC may significantly increase the risk of low-temperature cracking of those pavements. The problem of low-temperature cracks in asphalt layers of road pavements remains very important in many countries around the world despite climate change and related global warming. Additionally, due to weather anomalies, new regions of the world, where the phenomenon of low-temperature cracking had not occurred before, are increasingly exposed to the effects of considerably lower winter temperatures. Even in areas with an average temperature of −10 °C, HMAC should be used with caution [5]. After low-temperature cracks develop in the pavement, water enters the structure through the cracks. In consequence, pavement layers are weakened, and ride quality is limited. Therefore, in order to avoid premature deterioration of pavements, it is still necessary to investigate the relationships between the number of cracks and the low-temperature properties of asphalt layers. It is especially desirable to link properties obtained in laboratory conditions with the number of cracks observed in the field.

The mechanism of low-temperature crack formation is directly related to thermal tensile stresses in the asphalt layers, which are generated when temperature decreases. When the generated thermal stresses exceed the tensile strength of the layer, transverse cracking occurs. Prediction of thermal stress is very complex due to the viscoelastic nature of asphalt mixtures. Firstly, asphalt mixture stiffness increases with a decrease in temperature, which contributes to an increase in tensile stresses. Secondly, thermal stresses relax in time. A more detailed explanation of the mechanism of low-temperature crack formation, including thermal history and viscoelastic behavior of asphalt mixtures, was recently presented by Judycki [6,7,8]. The analysis of a viscoelastic model of asphalt layers showed that besides the value of extreme (minimum) temperature, the rate of temperature decrease and duration of low temperatures play a significant role as well [6,7,8]. Many studies like those performed by Hasan M.M. and Tarefder R.A. [9] concentrate on the identification of those properties of asphalt mixtures that improve the resistance to low-temperature cracking. The course of the construction stages and lack of suitable construction procedures can have a strong impact on pavement performance as well [3]. Therefore, the analysis of the obtained results should be performed, taking into consideration not only the laboratory tests but also information regarding the specific road section.

Thermal Stress Restrained Specimen Test (TSRST) is one of the most common test methods enabling the evaluation of low-temperature properties of asphalt mixtures. TSRST was introduced in 1965 by Monismith et al. [10]. In this method, a specimen is subjected to cooling with its ends restrained. Due to the limited movement of the specimen, thermal stress occurs that is similar to thermal loading in the field. The TSRST was further developed by Jung and Vinson as part of the Strategic Highway Research Program (SHRP) [11], and the influence of bitumen type, aging, and cooling rate on TSRST results was investigated. Many studies have indicated that the method has good sensitivity to low-temperature mixture design [12,13]. Additionally, Zaumanis et al. [14] stated that TSRST failure temperature variability was significantly smaller than the variability of Semi-Circular Bending (SCB) test fracture toughness, and TSRST distinguished different mixtures much clearer than the SCB test. Therefore, TSRST shows considerable advantages in mixture design and quality control applications. Many researchers utilize the TSRST method to assess the thermal properties of warm mix asphalt and Reclaimed Asphalt Pavement (RAP) content. Stienss et al. [15] studied the influence of selected warm mix asphalt additives on low-temperature properties of asphalt mixtures. Pham et al. [16] concluded that repeatability of the TSRST was good, based on the temperature and stress at failure. A decrease in air void content improved the low-temperature properties of asphalt mixtures with RAP. Izaks et al. [17] presented the TSRST results of HMAC containing high content of reclaimed asphalt material. It was concluded that conventional hard grade bitumen is not recommended for use in HMAC mixtures produced in regions with cold winters. In such regions, polymer-modified bitumens (PMB) are suitable for the improvement of low-temperature cracking resistance. Bankowski et al. [18] concluded that the use of highly-modified bitumens (HiMA) significantly improved the TSRST results of asphalt concrete. Badeli et al. [19] studied thermo-mechanical properties of asphalt base mixtures and their improvement obtained through the use of aramid pulp fibers. The TSRST test results showed that the addition of aramid fibers decreased the fracture temperature and stress at failure in comparison to the reference mixture. Studies conducted by Keshavarzi et al. [20] proved that the fracture stress and fracture temperature measured in TSRST could be predicted with reasonable accuracy using a dissipated pseudo strain energy-based failure criterion and the Simplified Viscoelastic Continuum Damage Model model. Zofka et al. [21] investigated three laboratory test methods (Indirect tension, semi-circular bending, and compact tension) that evaluate cracking resistance of asphalt mixtures at low temperatures on specimens obtained from 10 pavement sections in Minnesota and Illinois. However, limited research effort has been invested in the assessment of the TSRST results of specimens obtained from road sections in correlation with the observed cracking of those sections in real climatic conditions over a longer period of time.

### 1.2. Objectives

There are two principal objectives of the paper: (1) Assessment of low-temperature cracking behavior of high-modulus asphalt concrete layers in asphalt pavements; (2) Comparison of the laboratory TSRST test results of specimens collected in the field with observations of low-temperature cracks in trial sections.

## 2. Materials and Methods

### 2.1. Test Sections and Climatic Conditions

#### 2.1.1. Localization and Pavement Structures

Test sections are located near the city of Bialystok in the eastern part of Poland, in close distance to each other. The details of road sections, as well as basic data concerning particular asphalt layers, are summarized in Table 1. 

All sections were constructed under typical contract conditions as new flexible pavements. The oldest section was constructed in 2005 and the newest in 2012. Asphalt layers consist of three layers: wearing course of stone mastic asphalt (SMA), binder course, and asphalt base of high modulus asphalt concrete (HMAC). The maximum aggregate size in mm is given in the designation of the mixture. SMA mixtures were made using polymer-modified bitumens (PMB), in which styrene-butadiene-styrene was the modifier, while HMAC was made mostly using neat bitumen. In one section, the HMAC base and binder courses were made using styrene-butadiene-styrene modification. Designations of bitumens given in Table 1 are in accordance with the PN-EN 12591 and PN-EN 13808 standards and describe the penetration range and softening point (the latter in the case of PMB). The thickness of the wearing course equals 4 cm in each case, and the thicknesses of the remaining layers vary. Volumetric properties depend on the type of mixture (SMA or HMAC) and slightly vary on particular sections.

#### 2.1.2. Climatic Conditions

The performance grade for the region equals PG 52-28 on the level of wearing course and PG 46-28 on the level of binder course (at a reliability of 95%) [22]. Analyses of temperature conditions were performed on the basis of open-access data administrated by the Polish Institute of Meteorology and Water Management (IMGW) [23]. The data include daily measurements of air temperatures. Data on particular winter seasons from 1 December 2005 to 31 March 2021 were considered. Every winter season was counted as 4 months, from the beginning of December to the end of March. Figure 1 presents the minimum air temperature observed in each of the winter seasons. The temperature of pavement was calculated on the levels of the wearing course and the binder course using Equation (1), the same which is used to calculate the lower (minimum) temperature for Performance Grade system of bitumens [24]:(1)$$T_{min}^{d}=-1.56+0.72\cdot T_{air}-0.004\cdot \Phi^{2}+6.26\cdot \log_{10}\;\left(d+25\right).\;$$
where:

$T_{min}^{d}$: the minimum pavement temperature at depth d (°C);

$T_{air}\;$: the minimum annual air temperature (°C);

*Φ*: the latitude of the meteorological station (°);

d: the depth of the layer for which the temperature is calculated (mm), d=0. for the wearing course, and d=40 for the binder course.

**Figure 1 materials-15-00369-f001:**
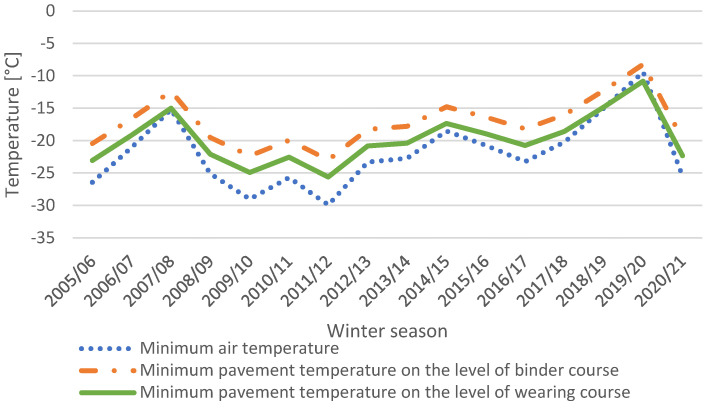
Minimum air and pavement temperatures during winter seasons for the considered sections.

On the basis of Figure 1 it can be stated that the extremely low air temperatures occurred in seasons 2009/10 and 2011/12, reaching as low as −26 °C (pavement) and −30 °C (air). Between 2014 and 2020, the minimum pavement temperature on the level of the wearing course equaled −21 °C. The minimum temperatures of pavement are higher than minimum air temperatures. Figure 1 also shows that minimum temperatures on the level of the binder course are higher than on the level of the wearing course, by 2 °C on average.

Apart from the minimum value of pavement temperature, the duration of low temperatures may also have an impact on the number of low-temperature cracks. When an asphalt mixture remains subjected to low temperature, the phenomena of physical hardening intensify [25,26]. In consequence, the stiffness modulus of the asphalt mixture increases, and thermal stresses induced in the asphalt layer may increase as well. However, the phenomenon is very complex, as thermal stresses relax in time. According to calculations performed by Judycki [3,13], the rate of temperature decrease has a significant effect on the increase in thermal stresses. Cyclic thermal changes also lead to thermal fatigue of asphalt mixtures [27] and, in consequence, contribute to the increase in the number of low-temperature cracks. Both physical hardening and thermal fatigue intensify with the increase in the number of days when air temperature remains low. Figure 2 summarizes the number of days on which air temperature fell below the following levels: −15 °C, −20 °C, and −25 °C. The freezing index calculated for particular winter seasons is presented as well. The freezing index used in this analysis is defined as the sum of the average daily temperatures from those days when the average daily temperature was below 0 °C—an increase in both statistics indicates a harsher winter season. 

As shown in Figure 1 and Figure 2, winter seasons from 2008 to 2012 were harsher than winter seasons after 2012. Some seasons, like 2019/20, were very mild. The total number of days when the temperature dropped below a given level forms the thermal history of the considered section. Thermal history is significant due to the fact that sections were constructed in various years between 2005 and 2012, as well as the fact that low-temperature cracks were investigated several times. 

### 2.2. Method of Field Assessment of Low-Temperature Cracks Intensity

The field investigation consisted of visual assessment of pavement distress, including cracks, ruts, roughness, and surface condition. For the analysis presented in this article, solely the information about low-temperature cracks was taken into account. It is noteworthy that for almost the entire length of the considered sections, transverse low-temperature cracks were the only visible form of distress. All the cracks which originated from causes other than low-temperature action were excluded from the analysis. The low-temperature cracks were clearly identified as single transverse cracks that were visible on the surface of each investigated section. Figure 3 presents examples of typical low-temperature transverse cracks occurring across the entire width of the carriageway, which was observed during the field investigation. In some rare cases, transverse cracks spanned only a portion of the width of the carriageway or were grouped as two or more cracks at a very low distance. In all the mentioned cases, they were counted as a single crack. Cracking index CI is defined as the average number of transverse cracks per 1 km of roadway. The field investigations were carried out in 2012, 2014, 2019, and 2020. Due to the fact that pavements were in various ages in the year of investigation, from 8 to 15 years old, the parameter of average Increase in Cracking index ICI was introduced. The ICI parameter characterizes the average number of new low-temperature cracks which appear after one winter season. It has to be mentioned that successive winter seasons contribute to the increase in the number of low-temperature cracks to a different extent; thus, the ICI parameter was determined for the period of several years.
(2)$$ICI_{Y}=\frac{CI_{Y}}{T}$$
where:

*ICI_Y_*: annual average increase in cracking index over the period from the year of construction to the year of investigation *Y*;

*CI_Y_*: cracking index of a given road section at the year of investigation *Y*;

*Y*: year(s) of investigation;

*T*: age of pavement (in years) at the moment of investigation.

**Figure 2 materials-15-00369-f002:**
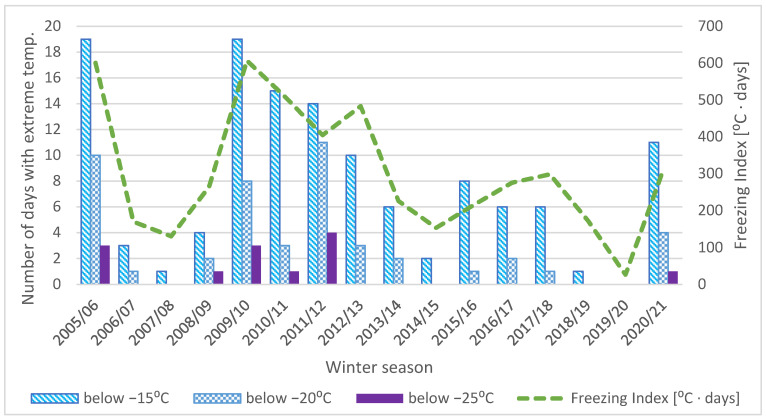
Freezing index and number of days when low air temperatures fell below the level of −15 °C, −20 °C, and −25 °C for the considered sections.

The results of the annual average increase in cracking index are summarized in Table 2. 

### 2.3. Method of Specimen Collection and Preparation for Laboratory Tests

Specimens for laboratory tests were drilled from each of the considered sections in the year 2018, and shortly after specimen collection, they were tested in the laboratory. The locations of specimen collection were chosen randomly. Two cores with a diameter of 300 mm were collected for each pavement layer: wearing course, binder course, and asphalt base. Prismatic specimens were subsequently sawn from the core samples and cut to the dimensions of 40 mm × 40 mm × 160 mm in the case of the wearing course and 50 mm × 50 mm × 160 mm in the case of the binder and base courses. The specimens were sawn from the central part of the core, parallel to the direction of traffic. The process of specimen collection and preparation is presented in Figure 4.

### 2.4. Laboratory Test Methods

The specimens collected from the field sections were tested in laboratory conditions using the (TSRST) according to the EN 12697-46 standard. In the TSRST, the specimen, whose length is held constant, is subjected to a decrease in temperature at a constant rate. Due to the prohibited thermal shrinkage, cryogenic (thermal) stress develops in the specimen. The results of the test include the progression of cryogenic stress over temperature *σ_cry_(T)* and failure stress *σ_cry_*, *_failure_(T)* at failure temperature *T_failure_*. Failure stress is equivalent to the strength of the specimen at failure temperature.

To perform the TSRST test, the following device was used TSRST—MULTI Multi-Station Thermal Asphalt System with servo-electric equipment (PAVETEST, Treviolo, Italy). The equipment and test setup used are presented in Figure 5.

In the TSRST procedure, the specimen is held at a constant length while the temperature is decreased at a uniform rate. The test starts at the temperature of T_0_ = +20 °C. For the standard test method, the cooling rate is set to 10 °C/h. The (cryogenic stress in the specimen gradually increases as the temperature decreases until the specimen fractures. The temperature at failure is the result of the test. The temperature-dependent cryogenic stresses *σ_cry_*(*T*) at −20 °C and at the temperature of failure are recorded as well.

The TSRST test has limitations in comparison to real conditions what is typical for any laboratory test. One of the most important limitations of the test is the fast cooling of the sample, with a gradient of 10 °C/h, while according to previous works [28], the gradient is lower than 3 °C/h. Specimens remain in low temperatures in a relatively short time (less than 3 h), while in real conditions, pavement remains several days or weeks at low temperatures. Thus the phenomena of physical hardening as well as the positive effect of thermal stress relaxation cannot be fully revealed [26]. Supplementing the research with physical hardening tests of asphalt mixtures [25] or binders [29] would give more comprehensive data on the mixtures, but it is beyond the scope of this work

## 3. Results and Discussion

### 3.1. Low-Temperature Cracks Intensity on the Basis of Field Investigation

Table 2 presents cracking indexes CI obtained for the considered road sections in successive years of investigation. On the basis of CI given in Table 2, the increases in cracking index ICI were calculated according to equation (2). In Figure 6, the values of ICI are presented in relation to the age of pavement in the year of investigation. The raw video records of tested sections are available in open access data set [30].

Figure 6a presents relations between the increase in cracking index and the age of pavement in the year of investigation. Each test section is represented by four points, which were obtained on the basis of investigations in particular years: 2012, 2014, 2019, and 2020. Each section was at a different age in the year of investigation and had a different history in terms of winter conditions. Winter seasons in successive years are not equal in terms of low temperatures, as shown in Section 2.1.2. The total number of days with extremely low temperatures, below −20 °C, counted over the entire period since the section was opened for service is proposed by the authors as a measure of the history of winter conditions. Figure 6b presents the relationship between the annual average increase in cracking index ICI and the total number of days on which air temperature decreased below −20 °C.

Figure 6a,b, and Table 2 show two trends: (1) number of new cracks decreases over time (see sections S8 Jezewo–Bialystok and DK8 Bialystok–Katrynka) or (2) number of new cracks slightly increases in the first years and then remains at a similar level (Sections DK19 Wasilkow beltway and DK8 Sztabin–Kolnica). After several years of service, the increase in the cracking index tends towards the mean value of all sections, which in the year 2020 equaled 0.57 new cracks per year. If the pavement shows a greater tendency to crack at the beginning of the service period, this tendency will be reduced in the further years of service and vice versa; if the pavement did not crack in the first years of service, it would tend to crack in further years. Such tendency is known in statistics as “reversion to the mean”. Several random factors affect the increase in cracking index in the first years of service; they are related to materials, the technology of asphalt layer paving, or work quality. For example, in the case of section S8 Jezewo–Bialystok, asphalt base and binder course were constructed in 2011, and those layers cracked during the first winter season (2011/12). All cracks were repaired and covered with a wearing course in 2012, but most of the cracks were reflected in the winter season 2012/13. Moreover, some new cracks occurred. In further years the increase in new cracks was much lower for this section. The observation may result from the fact that low-temperature cracks release tensile stresses during winter, and in consequence, the probability of occurrence of new cracks decreases. In contrast, section DK19 Wasilkow beltway was completed in 2011, including the wearing course, which may have protected the pavement against low-temperature cracks in the first winter seasons, but the cracks formed in further years. Therefore, the impact of the type of bitumen used decreases, while the impact of pavement age increases strongly. Analogously, section DK8 Sztabin–Kolnica, which is the oldest one, was subjected to the highest number of days with extremely low temperature–the increase in cracking index was lower at the beginning of service, and in the last seasons, it remained at the level of 0.4 new cracks per year.

The observations of the ICI factor in relation to pavement age and thermal history of pavements suggest that it is reasonable to consider ICI in the longest possible perspective. Such an approach will limit the influence of variability in the number of cracks in the first years of service, which results from random factors. In other words, “reversion to the mean” will be included. In further analysis, the authors compared ICI_2020_ of particular sections (obtained in the last year 2020) to the low-temperature performance of asphalt mixtures cored from respective sections.

### 3.2. Thermal Stress Restrained Specimen Test (TSRST) Results

The TSRST results (failure temperature, cryogenic stresses at failure and at −20 °C) of samples obtained from field sections are presented in Table 3, Table 4 and Table 5, separately for each asphalt layer. The coefficient of variation (CV) given in Table 3, Table 4 and Table 5 expresses the ratio between standard deviation and mean value of test results. The results are also presented graphically in Figure 7 and Figure 8.

Figure 7 presents the relationship between the maximum cryogenic stresses at failure and temperature at failure. The relationship between cryogenic stress at a temperature of −20 °C and at failure temperature is presented in Figure 8. Each point in Figure 7 and Figure 8 represents the mean obtained from three specimens per section and layer. The error bars shown in Figure 7 and Figure 8 represent standard deviations of results obtained for the three specimens.

The general tendency observed in Figure 7 is that when failure temperature is lower (meaning better resistance of the mixture to low-temperature cracking), the cryogenic stresses at failure are lower too. According to Figure 8, cryogenic stresses at failure are linearly related with cryogenic stresses at −20 °C. Cryogenic stresses increase with a decrease in temperature. These observations mean that mixtures with a higher capacity for stress relaxation are simultaneously more resistant to low-temperature cracking. HMAC mixtures, which are used for binder courses and asphalt bases, show higher cryogenic stresses at failure as well as higher (worse) failure temperatures than SMA mixtures for wearing courses. It is worth pointing out that HMAC is made from harder bitumen than SMA. SMA contains a higher amount of bitumen than HMAC. Interestingly, the results obtained in the TSRST test for asphalt concrete with 35/50 bitumen were similar to those obtained for asphalt concrete with harder 20/30 bitumen. One result, obtained for the wearing course of SMA with polymer-modified bitumen 45/80-65 on the section S8 Jezewo–Bialystok, deviates from the general trend significantly. The failure temperature is the lowest, and cryogenic stresses are the highest for this mixture. It suggests that the mixture has higher tensile strength than the remaining mixtures, which enables the mixture to withstand high cryogenic stresses. The mixture also has high relaxation capacity because cryogenic stresses at −20 °C are relatively low.

### 3.3. Relationships between TSRST and Pavement Low-Temperature Cracking Intensity

The relationships between TSRST results and pavement low-temperature cracking intensity ICI_2020_ are presented in Figure 9. Since ICI_2020_ includes reversion to the mean, it can be treated as a more reasonable parameter to express low-temperature cracking intensity than the cracking index, which is more susceptible to random factors like variations in minimum temperature in successive winter seasons or quality of section construction. Error bars shown in Figure 9 represent standard deviations of TSRST results analogously to Figure 7 and Figure 8. 

The following findings result from Figure 9:The lower the mean T_failure_ calculated for all three asphalt layers on a given section together, the lower the value of ICI_2020_ for the section, which means better resistance to low-temperature cracking of the whole pavement;It is important to consider TSRST results obtained for all asphalt layers, including the binder course and the asphalt base. The analysis of the following cases confirms this statement. Section DK8 Bialystok–Katrynka is characterized by the highest (the worst) values of T_failure_ for the wearing course (−25.2 °C) and relatively high values for the binder course (−23.4 °C) and the asphalt base (−21.5 °C). In consequence, the highest intensity of low-temperature cracking (expressed by the highest ICI_2020_ = 0.86) was observed in this section. In contrast, section S8 Jezewo–Bialystok is characterized by a slightly higher value of T_failure_ for the binder course (−21.5 °C) and the asphalt base (−20.8 °C), but significantly lower (much better) T_failure_ for the wearing course (−32.0 °C). Consequently, crack intensity observed on this section is low and ICI_2020_ = 0.41 new cracks per year;In approximation, a decrease in T_failure_ by 5 °C results in a decrease in the number of new cracks per year (ICI factor) by 0.45. It means that for a 15-year period, which is typically adopted as the period between reconstructions of wearing courses made from SMA, the number of low-temperature cracks can be reduced by approximately seven per each kilometer of the road;When asphalt mixtures show a greater capacity for relaxation of tensile stresses at low temperatures, sections are more resistant to low-temperature cracking. This observation results from Figure 9b. For sections where mixtures had lower cryogenic stresses at −20 °C, the average yearly increment of low-temperature crack intensity (ICI_2020_) decreased;The TSRST test has some limitations. Due to the fast cooling of specimens, the effects of physical hardening as well as stress relaxation have a minor impact on the final test results. Supplementing the research data by results of additional tests of physical hardening of asphalt mixtures could result in a better relationship between laboratory test results and field observations of low-temperature crack intensity.

The performed laboratory tests and observations of field sections provide some implications for pavement design. When the use of HMAC is planned (which may result in worse resistance to low-temperature cracking than conventional AC, measured, e.g., by T_failure_ in TSRST), it is advisable to apply a wearing course with improved resistance to low-temperature cracking and lower value of T_failure_. Application of polymer-modified bitumen with higher content of modifier, e.g., PMB 45/80-65 instead of PMB 45/80-55, can be one of the possible solutions. 

## 4. Summary and Conclusions

Four test sections with HMAC mixture used in the base and binder courses and SMA mixture used in the wearing course were evaluated in this study. Sections are localized in the same climatic zone in the northeast of Poland, where Performance Grade for wearing course equals PG 52-28. Sections were constructed under typical contractor conditions and have been in service longer than nine years. Based on the test results and analysis, the following conclusions can be drawn:It was observed that when a given pavement showed a greater tendency to crack at the beginning of the service period, this tendency was reduced in the further years of service and vice versa if the pavement did not crack in the first years of service, it tended to crack in the further years. It means that the number of new low-temperature cracks is susceptible to many random factors, and reversion to the mean should be considered for an appropriate assessment of the resistance of pavements to low-temperature cracking. Therefore, the new factor named Increase in Cracking Index was proposed, expressing the average annual increase in the number of low-temperature cracks per kilometer in long-term perspective;For each of the considered sections, samples of every asphalt layer were collected, and TSRST was performed and analyzed. The observed general tendency was that when failure temperature is lower (which means better resistance of the mixture to low-temperature cracking), the cryogenic stresses at failure are lower too. One case of SMA mixture with polymer-modified bitumen 45/80-65 deviates from the general trend significantly and suggests that the mixture has much higher tensile strength than the remaining mixtures, which enables the mixture to withstand high cryogenic stresses;Correlations of temperature at failure and cryogenic stresses in TSRST with cracking intensity observed in the field were analyzed. The lower the mean T_failure_ calculated together for all three asphalt layers on a given section, the lower the value of ICI_2020_ on this section, which means better resistance to low-temperature cracking of the whole pavement. It was also observed that when asphalt mixtures show a greater capacity for relaxation of tensile stresses at low temperatures, sections are more resistant to low-temperature cracking;The paper provides a practical suggestion for pavement designers. When the use of HMAC for the binder course and the asphalt base is planned, which may result in worse resistance to low-temperature cracking of pavement than in the case of usage of conventional asphalt concrete, it is advisable to apply for a wearing course with improved resistance to low-temperature cracking. Such an approach may compensate for the adverse effects of usage of HMAC.

## Figures and Tables

**Figure 3 materials-15-00369-f003:**
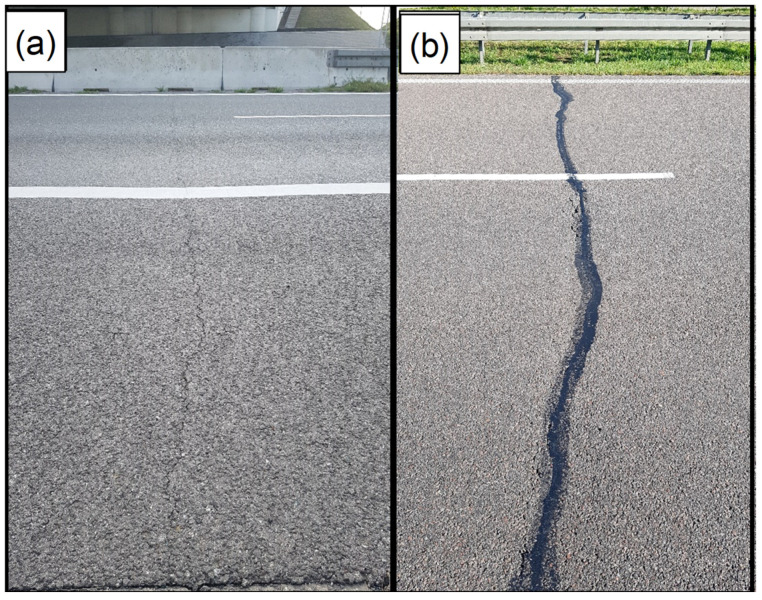
Examples of low-temperature cracks. (**a**) Unrepaired; (**b**) Sealed (Expressway S8).

**Figure 4 materials-15-00369-f004:**
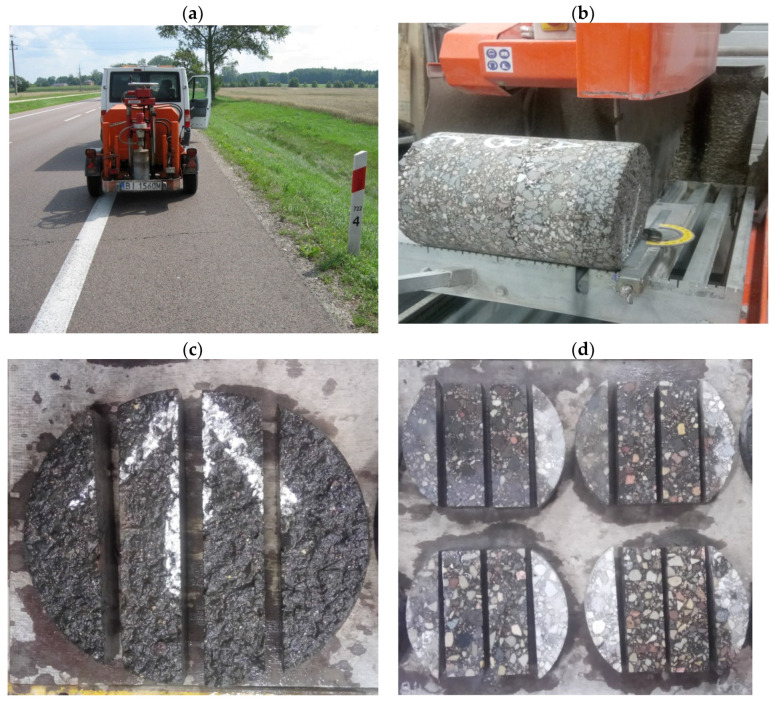
The method of specimen preparation: (**a**) core specimens drilled from road sections, (**b**) division of cores into individual layers, (**c**) cutting along the direction of vehicle movement, and (**d**) formation of prismatic beams.

**Figure 5 materials-15-00369-f005:**
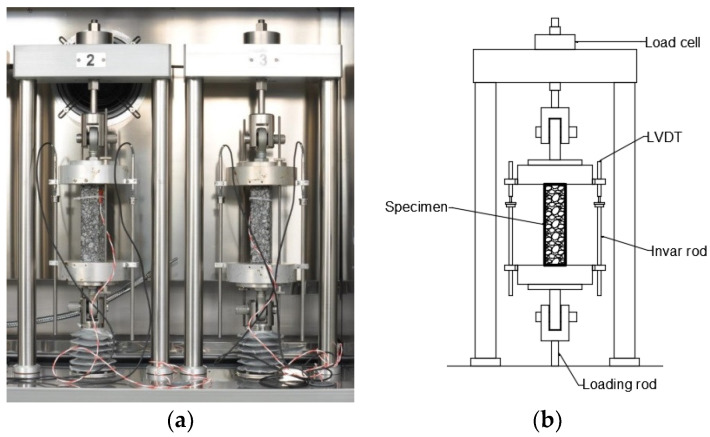
Thermal stress restrained specimen test (TSRST) setup. (**a**) Photograph of specimens during the test; (**b**) Schematic view [15].

**Figure 6 materials-15-00369-f006:**
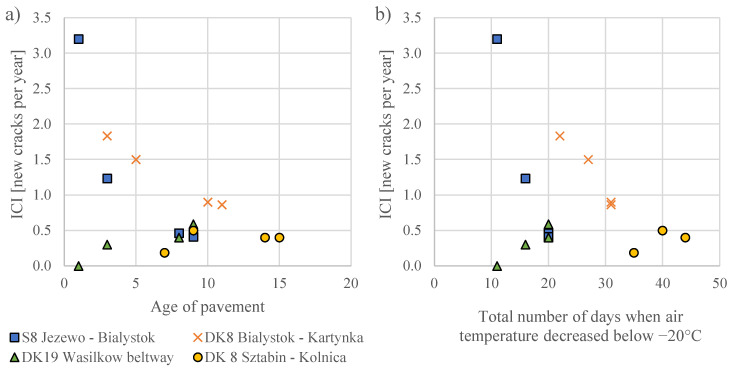
The annual average increase in cracking index in relation to (**a**) pavement age and (**b**) total number of days when air temperature decreased below −20 °C.

**Figure 7 materials-15-00369-f007:**
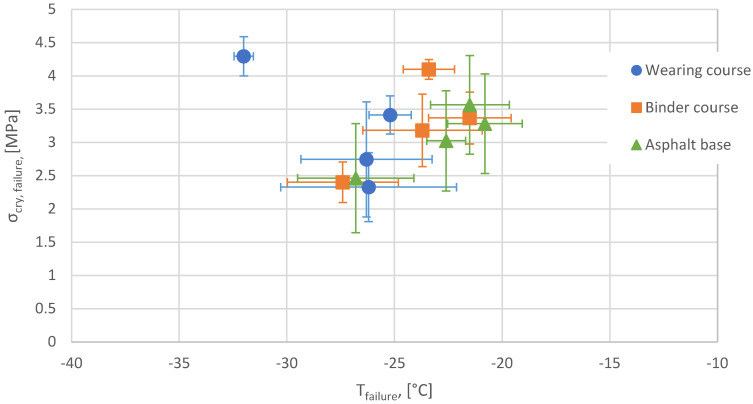
The relationships between cryogenic stresses at failure and temperature of failure in the TSRST test.

**Figure 8 materials-15-00369-f008:**
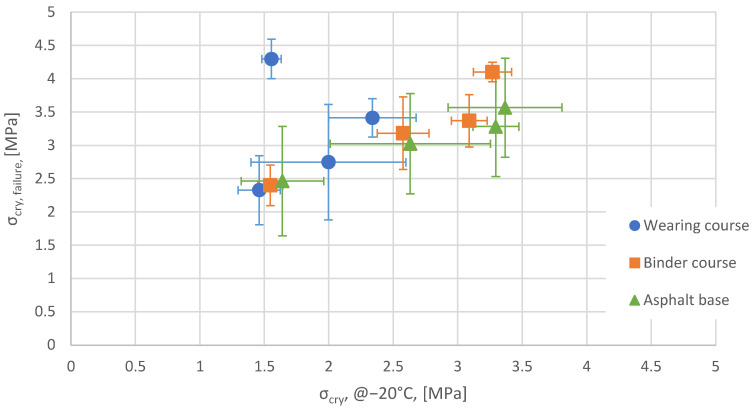
The relationships between cryogenic stresses at −20 °C and cryogenic stresses at failure temperature.

**Figure 9 materials-15-00369-f009:**
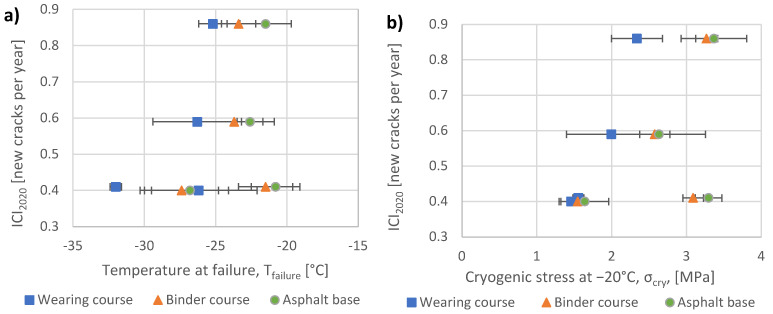
Relations between the increase in cracking index obtained for the period from the year of pavement construction up to 2020 (ICI_2020_) and the results of the TSRST test. (**a**) Failure temperature; (**b**) Cryogenic stress at −20 °C.

**Table 1 materials-15-00369-t001:** Asphalt courses of the analyzed pavements.

Road Number and Section Description	Year of Construction	Asphalt Layer	Mixture Type, Aggregate Gradation, and Type of Bitumen	Thickness (cm)	Voids (%)	Bitumen (% m-m)	Density (g/cm^3^)
S8 Jezewo–Bialystok km 614 + 850 to 639 + 365 Expressway	2012	Wearing course	SMA 11 PMB 45/80-65	4	3.2	6.6	2.336
		Binder course Asphalt base	HMAC 16 20/30	8	2.7	5.0	2.409
			HMAC 16 20/30	16	2.5	5.1	2.415
S8 Bialystok–Katrynka km 648 + 117 to 654 + 548 Expressway	2009	Wearing course	SMA 11.2 PMB 45/80-55	4	3.6	6.5	2.335
		Binder course	HMAC 16 20/30	8	3.0	4.8	n/a
		Asphalt base	HMAC 16 20/30	16	3.0	4.8	n/a
DK19 Wasilkow beltway km 45 + 700 to 50 + 700 National Road	2011	Wearing course	SMA 11.2 PMB 45/80-55	4	3.6	6.5	2.335
		Binder course	HMAC 20 35/50	6	5.2	4.3	2.376
		Asphalt base	HMAC 25 35/50	8	5.6	4.0	2.378
DK8 Sztabin–Kolnica km 717 + 982 to 723 + 236 National Road	2005	Wearing course	SMA 12.8 PMB 45/80-55	4	n/a	n/a	n/a
		Binder course	HMAC 20 10/40-65	8	3.2	4.8	2.400
		Asphalt base	HMAC 20 10/40-65	9	3.2	4.7	2.401

**Table 2 materials-15-00369-t002:** Results of the field investigations of trial sections.

Road Section	Year of Construction	Cracking Index CI				Increase in Cracking Index ICI			
		CI_2012_	CI_2014_	CI_2019_	CI_2020_	ICI_2012_	ICI_2014_	ICI_2019_	ICI_2020_
S8 Jezewo–Bialystok	2011	3.2	3.5	3.7	3.7	3.20	1.23	0.46	0.41
DK8 Bialystok–Katrynka	2009	5.5	8.7	9.5	9.5	1.83	1.50	0.90	0.86
DK19 Wasilkow beltway	2011	0.0	1.3	3.2	5.3	0.00	0.30	0.40	0.59
DK 8 Sztabin–Kolnica	2005	1.3	5.0	5.3	6.0	0.19	0.50	0.40	0.40

**Table 3 materials-15-00369-t003:** Test results of TSRST for the wearing course.

Road Section		σ_cry, failure_ (MPa)	T_failure_ (°C)	σ_cry, @−20°C_ (MPa)
S8 Jezewo–Bialystok	mean value	4.296	−32.0	1.555
	CV (%)	6.9	1.4	4.8
DK8 Bialystok–Katrynka	mean value	3.413	−25.2	2.338
	CV (%)	8.4	3.9	14.5
DK19 Wasilkow beltway	mean value	2.747	−26.3	1.997
	CV (%)	31.5	11.6	30.1
DK 8 Sztabin–Kolnica	mean value	2.328	−26.2	1.460
	CV (%)	22.3	15.6	11.2

**Table 4 materials-15-00369-t004:** Test results of TSRST for the binder course.

Road Section		σ_cry, failure_ (MPa)	T_failure_ (°C)	σ_cry, @−20°C_ (MPa)
S8 Jezewo–Bialystok	mean value	3.368	−21.5	3.089
	CV (%)	11.6	8.9	4.5
DK8 Bialystok–Katrynka	mean value	4.100	−23.4	3.269
	CV (%)	3.6	5.1	4.5
DK19 Wasilkow beltway	mean value	3.183	−23.7	2.576
	CV (%)	17.1	11.7	7.8
DK 8 Sztabin–Kolnica	mean value	2.403	−27.4	1.546
	CV (%)	12.7	9.4	6.3

**Table 5 materials-15-00369-t005:** Test results of TSRST for the base course.

Road Section		σ_cry, failure_ (MPa)	T_failure_ (°C)	σ_cry, @−20°C_ (MPa)
S8 Jezewo–Bialystok	mean value	3.282	−20.8	3.295
	CV (%)	22.8	8.3	5.4
DK8 Bialystok–Katrynka	mean value	3.566	−21.5	3.366
	CV (%)	20.8	8.5	13.1
DK19 Wasilkow beltway	mean value	3.025	−22.6	2.632
	CV (%)	24.9	4.0	23.6
DK 8 Sztabin–Kolnica	mean value	2.464	−26.8	1.640
	CV (%)	33.3	10.1	19.5

## Data Availability

Data available in a publicly accessible repository The data presented in this study are openly available in repository: *Investigation of Low-Temperature Cracks on Selected National Roads and Motorways in Poland 2020, Bridge of Data.* Gdansk University of Technology at doi: 10.34808/an8a-3k90.

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
