# Peer review of "Field Evaluation of High Modulus Asphalt Concrete Resistance to Low-Temperature Cracking"

_materials, 2022, doi:10.3390/ma15010369_

Round 1
Reviewer 1 Report
High-modulus asphalt concrete (HMAC) has numerous advantages in comparison to conventional asphalt concrete and is widely used in pavement construction. In this study, the low-temperature cracking resistance of HMAC was evaluated under different service years and the number of low-temperature cracks was investigated in the field and the resistance of pavement to low-temperature cracking was tested using Thermal Stress Restrained Specimen Tests (TSRST). Furthermore, the correlations of temperature at failure and cryogenic stresses in TSRST with the cracking intensity observed in the field were analyzed. This study is interesting and relevant to practice, which can be considered after a major revision. Some comments in below can be considered in this paper.
- 1. Background, the “past studies” may be more appropriate to change to “previous studies”.
- 3. “Method of specimen collection and preparation for laboratory tests”, Please explain why the locations of specimen collection were chosen randomly, whether to consider the impact of different lane sampling on the test results.
- Are the two cores drilled from the same location for each pavement layer or a different location? Because the adjacent position sampling maximizes the consistency of the material and reduces the impact of material differences on the test results.
- 2. “Thermal stress restrained specimen test (TSRST) results”, the time of specimen obtained from field sections should be supplemented, 2020 or other years.
- Since only two core samples were obtained for each section and the four test samples were prepared (Figure 4(d)), however, Figures 7 and 8 represents the mean obtained from three specimens per each section and each layer. The number of samples that can be obtained by coring is not consistent with the number of test samples.
Author Response
Answers to the comments and suggestions:
Regarding the detailed suggestions, our answers are as follows:
General comment: High-modulus asphalt concrete (HMAC) has numerous advantages in comparison to conventional asphalt concrete and is widely used in pavement construction. In this study, the low-temperature cracking resistance of HMAC was evaluated under different service years and the number of low-temperature cracks was investigated in the field and the resistance of pavement to low-temperature cracking was tested using Thermal Stress Restrained Specimen Tests (TSRST). Furthermore, the correlations of temperature at failure and cryogenic stresses in TSRST with the cracking intensity observed in the field were analyzed. This study is interesting and relevant to practice, which can be considered after a major revision.
Answer to comment: The authors would like to thank the reviewer for the comment.
Comment 1: Background, the “past studies” may be more appropriate to change to “previous studies.
Answer to 1: It was corrected in the paper.
Comment 2: Method of specimen collection and preparation for laboratory tests”, Please explain why the locations of specimen collection were chosen randomly, whether to consider the impact of different lane sampling on the test results.
Answer to 2: The locations of specimen collection were chosen randomly according to the length of each road section and the method of how specimens were drilled from pavement was the same at each section.
Comment 3: Are the two cores drilled from the same location for each pavement layer or a different location? Because the adjacent position sampling maximizes the consistency of the material and reduces the impact of material differences on the test results.
Answer to 3: Yes, two or sometimes even three cores were drilled for each pavement layer at the single location. The specimens were drilled close to each other. The influence of material differences on the test results is limited.
Comment 4: 2. “Thermal stress restrained specimen test (TSRST) results”, the time of specimen obtained from field sections should be supplemented, 2020 or other years.
Answer to 4: It was supplemented in the paper (in Par. 2.3): “Specimens for laboratory tests were drilled from each of the considered sections in year 2018 and shortly after specimen collection they were tested in the laboratory.”
Comment 5: Since only two core samples were obtained for each section and the four test samples were prepared (Figure 4(d)), however, Figures 7 and 8 represents the mean obtained from three specimens per each section and each layer. The number of samples that can be obtained by coring is not consistent with the number of test samples.”
Answer to 5: According to statistical assessment of the TSRST test results the minimum number of specimens tested was 3.
Reviewer 2 Report
The research topic is actual and relevant to the Special Issue. The overall investigation procedure is described properly.
A few minor comments
- page 2
“Course of the construction stages and lack of suitable construction procedures can have a strong impact on pavement performance as well”
please rephrase
- page 2
“Due to limited movement of the specimen, thermal stress occurs that is similar to thermal loading in the field”
please rephrase
- page 2
“PMB binders are suitable”
Please remove the extra space
Author Response
Answers to the comments and suggestions:
Regarding the detailed suggestions, our answers are as follows:
General comment: The research topic is actual and relevant to the Special Issue. The overall investigation procedure is described properly.
Answer to comment: The authors would like to thank the reviewer for the comment.
Comment 1: “Course of the construction stages and lack of suitable construction procedures can have a strong impact on pavement performance as well.” Please rephrase.
Answer to 1: That statement that was used in the paper results from the research conducted. The conclusions were published in 2015 (see ref. [3]).
Comment 2: “Due to limited movement of the specimen, thermal stress occurs that is similar to thermal loading in the field”. Please rephrase.
Answer to 2: The statement describes the TSRST test method applied in the paper. According to limited movement of the specimen that method is suitable to simulate thermal loading at laboratory.
Comment 3: “PMB binders are suitable”. Please remove the extra space”.
Answer to 3: It was corrected in the paper.
Reviewer 3 Report
Interesting paper and the results are well presented. Low-temperature cracking is a common distresses mode of asphalt pavement in cold northern regions. One of the main merits of this paper is that the author used asphalt materials obtained in field instead of indoor laboratory aging simulation materials for testing. The data obtained from the tests performed can more truly reflect the performance of the on-site materials. It is worth mentioning that TSRST method has some disadvantages such as cannot detect microcracks formed and cannot apply very low cooling rate. The authors are recommened to discuss more about the method used. In addition, in order to avoid stress concentration, cylindrical specimens are usually used for testing instead of rectangular parallelepiped. The authors should discuss more about the test configuration selected. Last but not least, it is recommended that the authors consider the thermoreversible aging phenomenon in the asphalt mixture characterization, which may not easily detected by mechanical tests.
Author Response
Answer to comments:
The authors would like to thank the reviewer for the comments. The authors agree with the reviewer that TSRST method has some disadvantages. As many laboratory test methods, also TSRST method was developed to represent the field conditions as close as it can be possible. In that method the detection of microcracks formed in a specimen is impossible. But microcracks can not be also observed in the field. In TSRST method application of very low cooling rate is possible and was also presented in the paper (10.1016/j.conbuildmat.2019.01.148). In our opinion TSRST method is one of the most suitable method to assess the field sections according to mechanism of low temperature cracking formation and because that method is commonly used around the world. The laboratory tests using TSRST method were performed according to EN 12697-46 standard that allows to use rectangular specimens. The same type of specimens is also use by other research groups in Europe and other countries (for example in Austria, Germany, etc.). According to thermoreversible aging phenomenon (physical hardening) the authors agree with the reviewer that it may not be easily detected by mechanical tests. The main objective of the paper is comparison of the laboratory TSRST test results of specimens collected in field with observations of low-temperature cracks in real conditions. So it could be difficult to discuss all issues mentioned in the review.
Reviewer 4 Report
Field Evaluation of Resistance of High-Modulus Asphalt Concrete to Low-Temperature Cracking
This paper is addressing an essential problem in the HMAC which is low-temperature cracking with introducing a new factor named as cracking index using thermal stress restrained specimen test which looks interesting. The objective and problem statement looks well written and explained.
It needs to be proofread and reference checked. This paper needs serious editing it is not well written and it is hard to follow sometimes (seems like I has been written in hurry).
-Page 1 line 12: author can avoid acronyms in the abstract (ex. TSRST).
-Referencing needs serious attention. Citations and references are not standard according to any of the usual methods. (ex. Page 1 line 21, Chen et al. [1], Xu et al. [2]??)
-Page 2 line 6: References?
-Page 2 line 15: References?
-Page 2 line 21: Acronym only in one place and don’t repeat.
-Page 2 line 25: Strategic highway research program capitalized?
-Page 2 line 28: SCB stands for (define it first)?
-Page 2 line 30: RAP stands for (define it first)?
-Page 2 line 30: PMB, HiMA (highly-modified binder??) stands for (define it first)?
-Page 3: S-VECD? IDT? DC(T)?
-Page 3: Please elaborate more on the objective section with more information
-Page 3: Wearing course, binder course, asphalt base: better naming is suggested.
-Please be consistent with using “asphalt”, “binder”, and “bitumen”
-Page 5: How freezing index is calculated. Reference?
-Figure 4 would be better if all pictures were in same size and organized
-Section 2.4: TSRST acronym was defined again
-Section 2.4: Cryogenic (thermal) stress used twice. After first “(thermal)” you don’t need to repeat it again
-Section 2.4: “The specimens were tested using TSRST—MULTI Multi-Station Thermal Asphalt System servo
electric equipment (PAVETEST, Italy).” Rewrite the sentence
-Page 8: “The thermally-induced (cryogenic) stress in the specimen gradually increases as the temperature decreases, until the specimen fractures. The temperature at failure is the result of the test. The temperature-dependent thermal (cryogenic) stresses σcry(T) at -20°C and at temperature of failure are recorded as well.” No need to use cryogenic and thermal both each time.
-Results and discussion: use either “cracking index” or “CI” if you already defined. “Table 2 presents cracking indexes CI obtained for the considered road sections in successive years of investigation.”
-Figure 6b: specific year for each bullet point can be identified on the each one.
-Figure 6b: Why DK 8 Sztabin – Kolnica has only 3 bullet points?
-Figure 6: I would suggest to find a better illustration in order to see the ICI in the years of investigation as well. It is hard to follow the discussion with the figure 6.
-“(Sections DK19 Wasilkow beltway and SK8 Sztabin–Kolnica).” SK8 Sztabin-Kolnica or DK8?
-I suggest to separate the Figure 6a and 6b and discuss the separately.
-Figure 7 and 8: I suggest to identify each section as well. You can use shapes for layer and color for the road sections
-Figure 9, vertical axis: apostrophe was used instead of dot (other figures have same problem as well).
-Figure 9a and b can be separate and discussed separate as well. Better to identify the sections too.
-Page 12: no need to number the results from figure 9.
-Conclusion: bullet points will help to organize this section rather than writing in paragraphs.
-“Four test sections with HMAC mixture used in the base and binder courses and SMA mixture used in the wearing course were evaluated in this study. Sections are localized in the same climatic zone in the north-east of Poland, where Performance Grade for wearing course equals PG 52-28. Sections were constructed under typical contractor conditions and have been in service longer than 9 years.” Not a conclusion!
-Conclusion section looks like a discussion and result. I suggest to use short bullet points.
-Page 13: “For each of the considered sections, samples of every asphalt layer were collected and Thermal Stress Restrained Specimen Tests (TSRST) were performed and analyzed.” Not a conclusion.
Author Response
Reviewer remark: -Page 1 line 12: author can avoid acronyms in the abstract (ex. TSRST).
Authors answer: The change was introduced.
Reviewer remark: -Referencing needs serious attention. Citations and references are not standard according to any of the usual methods. (ex. Page 1 line 21, Chen et al. [1], Xu et al. [2]??)
Authors answer: Referencese were corected according to the remark.
Reviewer remark: -Page 2 line 6: References?
Authors answer: This is a general statement, no special studies refer to this observation.
Reviewer remark: -Page 2 line 15: References?
Authors answer: The sentence is a summary of studies peerfomed by Judycki [6,7,8]
Reviewer remark: -Page 2 line 21: Acronym only in one place and don’t repeat.
Authors answer: It was corrected,.
Reviewer remark: -Page 2 line 25: Strategic highway research program capitalized?
Authors answer: It was corrected
Reviewer remark: -Page 2 line 28: SCB stands for (define it first)?
Authors answer: Semi-Circular Bending – we made a correction
Reviewer remark: -Page 2 line 30: RAP stands for (define it first)?
Authors answer: Reclaimed Asphalt Pavement –the correction was introduced
Reviewer remark: -Page 2 line 30: PMB, HiMA (highly-modified binder??) stands for (define it first)?
Authors answer: The detailed description was introduced
Reviewer remark: -Page 3: S-VECD? IDT? DC(T)?
Authors answer: full desription was added
Reviewer remark: -Page 3: Please elaborate more on the objective section with more information
Authors answer: We modified objectives to make in clearer for a wider audience.
Reviewer remark: -Page 3: Wearing course, binder course, asphalt base: better naming is suggested.
Authors answer: Names are appropriate. There are no better names for this layers.
Reviewer remark: -Please be consistent with using “asphalt”, “binder”, and “bitumen”
Authors answer: We checked the names and corrected “binder” to “bitumen” to be more consistent
Reviewer remark: -Page 5: How freezing index is calculated. Reference?
Authors answer: There are several approachces in calculation of freezing index. We used the following approach: Freezing index used in this analysis is defined as the sum of the average daily temperatures from those days when the average daily temperature was below 0°C., what is given in the text.
Reviewer remark: -Figure 4 would be better if all pictures were in same size and organized
Authors answer: Figure 4 was reorganized.
Reviewer remark: -Section 2.4: TSRST acronym was defined again
Authors answer: It was corrected
Reviewer remark: -Section 2.4: Cryogenic (thermal) stress used twice. After first “(thermal)” you don’t need to repeat it again
Authors answer: It was corrected
Reviewer remark: -Section 2.4: “The specimens were tested using TSRST—MULTI Multi-Station Thermal Asphalt System servo
Reviewer remark: electric equipment (PAVETEST, Italy).” Rewrite the sentence
Authors answer: Both remarks were corrected
Reviewer remark: -Page 8: “The thermally-induced (cryogenic) stress in the specimen gradually increases as the temperature decreases, until the specimen fractures. The temperature at failure is the result of the test. The temperature-dependent thermal (cryogenic) stresses σcry(T) at -20°C and at temperature of failure are recorded as well.” No need to use cryogenic and thermal both each time.
Authors answer: It was corrected
Reviewer remark: -Results and discussion: use either “cracking index” or “CI” if you already defined. “Table 2 presents cracking indexes CI obtained for the considered road sections in successive years of investigation.”
Authors answer: We prefer to use both in the titles of figures and tables. Not each reader read whole paper.
Reviewer remark: -Figure 6b: specific year for each bullet point can be identified on the each one.
Authors answer: Points on the right side of chart represents measurements from later years.
Reviewer remark: -Figure 6b: Why DK 8 Sztabin – Kolnica has only 3 bullet points?
Authors answer: There are 4, but points from 2019 and 2020 lies on the same position, as you can find in table 2.
Reviewer remark: -Figure 6: I would suggest to find a better illustration in order to see the ICI in the years of investigation as well. It is hard to follow the discussion with the figure 6.
Authors answer: In discussion it is better to consider both table 2 and figure 6. Some additional figure as it is recommended by reviewer would be duplication of table 2.
Reviewer remark: -“(Sections DK19 Wasilkow beltway and SK8 Sztabin–Kolnica).” SK8 Sztabin-Kolnica or DK8?
Authors answer: DK8, it was an Editorial mistake
Reviewer remark: -I suggest to separate the Figure 6a and 6b and discuss the separately.
Authors answer: In the author’s opinion it is better to consider figures 6a and 6b together because both charts shows changes of ICI in relation to time: age or total number of days when air temperature decreased below -20.
Reviewer remark: -Figure 7 and 8: I suggest to identify each section as well. You can use shapes for layer and colour for the road sections
Authors answer: The details about results for particular sections are presented in tables 3-5. The purpose offigures 7 and 8 is to present relationships. Too many information at one figure may complicate and obscure the main thought
Reviewer remark: -Figure 9, vertical axis: apostrophe was used instead of dot (other figures have same problem as well).
Authors answer: It was corrected. Dots should be used on each values
Reviewer remark: -Figure 9a and b can be separate and discussed separate as well. Better to identify the sections too.
Authors answer: Similar as in previous remark. In the author’s opinion it is better to consider figures 9a and 9b together because both charts shows changes of ICI in relation to results of TSRST test:
Reviewer remark: -Page 12: no need to number the results from figure 9.
Authors answer: It was corrected.
Reviewer remark: -Conclusion: bullet points will help to organize this section rather than writing in paragraphs.
Authors answer: It was corrected.
Reviewer remark: -“Four test sections with HMAC mixture used in the base and binder courses and SMA mixture used in the wearing course were evaluated in this study. Sections are localized in the same climatic zone in the north-east of Poland, where Performance Grade for wearing course equals PG 52-28. Sections were constructed under typical contractor conditions and have been in service longer than 9 years.” Not a conclusion!
Authors answer: The title of the section was changed into: summary and conclusions
Reviewer remark: -Conclusion section looks like a discussion and result. I suggest to use short bullet points.
Authors answer: It was corrected.
Reviewer remark: -Page 13: “For each of the considered sections, samples of every asphalt layer were collected and Thermal Stress Restrained Specimen Tests (TSRST) were performed and analyzed.” Not a conclusion.”
Authors answer: This is a summary. The title of section was changed.
Reviewer 5 Report
Thank you for submitting your work.
This is an interesting laboratory and in situ evaluation on the resistance of different HMAs to low-temperature cracking. The paper is well written and easy to follow and the analysis of results is in line with the values obtained from the laboratory characterization. I have nothing to add on the results, but I would like to know if the authors have some information about the traffic conditions of the tested pavements. Do you think that the traffic could have an influence on the formation of the low-temperature cracking?
Author Response
Answer to comments:
The authors would like to thank the reviewer for the comments. The traffic conditions have not being assessed. All road sections presented in the paper characterize the similar traffic conditions. In our opinion traffic conditions could not have direct influence on the formation of the low-temperature cracking.
Round 2
Reviewer 1 Report
All my comments are well-addressed, I am okay to see this work to be published on the journal. Thank you for your response.
Author Response
Thank you so much.